

# An assessment of Ozone Mini-holes Representation in Reanalyses Over the Northern Hemisphere

Luis F. Millán[1] and Gloria L. Manney[2,3]

[1]Jet Propulsion Laboratory, California Institute of Technology, Pasadena, California, USA
[2]NorthWest Research Associates, Socorro, New Mexico, USA
[3]New Mexico Institute of Mining and Technology, Socorro, New Mexico, USA

*Correspondence to:* L. Millán (luis.f.millan@jpl.nasa.gov)

**Abstract.** An ozone mini-hole is a synoptic-scale region with strongly decreased total column ozone resulting from dynamical processes. Using total column measurements from the Ozone Monitoring Instrument and ozone profile measurements from the Microwave Limb Sounder, we evaluate the accuracy of mini-hole representation in five reanalyses. This study provides a metric of the reanalyses' ability to capture dynamically-driven ozone variability. The reanalyses and the measurements show similar

seasonal variability and geographical distributions of mini-holes; however, all of the reanalyses underestimate the number of mini-holes, their area, and in many reanalyses their location displays an eastward bias. The reanalyses' underestimation of mini-hole number ranges from about 34% to about 83%. The mini-hole vertical representation in the reanalyses agrees well with that in the MLS measurements and, furthermore, is consistent with previously reported mechanisms for mini-hole formation. The skill of the reanalyses is not closely tied to the ozone fields assimilated, suggesting that the dynamics of the

reanalysis models are more important than the assimilated ozone fields to reproducing ozone mini-holes.

## 1 Introduction

Since early ozone measurements (Dobson et al., 1929), it has been known that the total column ozone is characterized by day-to-day local fluctuations that are correlated with the passing of synoptic weather systems. Meetham and Dobson (1937)

found a significant correlation between the total column ozone and stratospheric temperature and potential temperature, as well as the density and height of the tropopause. Reed (1950) studied the relative importance of horizontal advection and vertical motion to producing such fluctuations and the manner in which those two mechanisms combine to produce the ozone-weather relationships.

Events with very low total ozone columns were named ozone mini-holes by Newman et al. (1988) because of their rapid

ozone decline and their synoptic scale, which was in contrast to the slow ozone decline and planetary scale of the Antarctic ozone hole. These events are found mainly throughout mid-latitudes on both hemispheres (e.g., James, 1998a, b; Hood et al., 2001). Unlike the well-known Antarctic ozone hole, mini-holes are mainly driven by dynamical-atmospheric processes rather than photo-chemical processes. As pointed out by Reed (1950), they result from a combination of uplift of air and horizontal advection. Local uplift of the air increases the amount of the column occupied by ozone-poor tropospheric air (e.g., Salby





and Callaghan, 1993; Petzoldt et al., 1994; Teitelbaum et al., 1998; Hood et al., 2001; Canziani et al., 2002) while horizontal advection brings ozone-poor air into the column through poleward transport around the tropopause, through equatorward transport of polar air around the middle of the stratosphere, or a combination of the two (e.g., Vaughan and Price, 1991; Orsolini et al., 1995; Peters et al., 1995; James et al., 2000; Allen and Nakamura, 2002; James and Peters, 2002; Mangold
et al., 2009).

As an example, Figure 1 shows the mini-hole event observed over the UK on 19 January 2006. On this day, a record low total ozone column of 177 DU was observed at Reading (normally is around 310 DU in this time of the year) as well as low total ozone columns over other Northwest European stations (Keil et al., 2007). Figure 1 also shows the same region a few days before an after the event to illustrate the transient nature of this phenomenon.

Reanalyses from data assimilation systems provide globally-gridded high resolution meteorological fields based on a optimized combination of general circulation models and observational data. Reanalyses from the latest generation also provide fields of assimilated ozone, but use different ozone inputs and assimilation methods (e.g., Fujiwara et al., 2017). In this paper, the accuracy of mini-hole representation is used as a metric to assess the reanalyses' ability to capture dynamically-driven ozone variability. We used five of the most recent high-resolution reanalyses: the European Centre for Medium-Range Weather
Forecasts (ECMWF) re-analysis (ERA) Interim, the National Centers for Environmental Prediction Climate Forecast System Reanalysis (CFSR), the Japanese 55-year Reanalysis (JRA-55), the Modern-Era Retrospective Analysis for Research and Applications (MERRA) and its successor MERRA-2. We evaluate the mini-hole representation comparing the mini-hole events' area, distance, and orientation with respect to the events found in the Ozone Monitoring Instrument (OMI) data (Levelt et al., 2006). Further, we use vertical profiles of ozone and temperature from the Aura Microwave Limb Sounder (MLS) (Waters
et al., 2006), as well as derived meteorological products (DMPs) (Manney et al., 2007) and detailed tropopause information from the JEt and Tropopause Products for Analysis and Characterization (JETPAC) package (Manney et al., 2011), to study the vertical structure and relationships to the tropopause of the mini-holes in reanalyses and satellite data.

## 2  Data

In this section a description is given of the observational data as well as the reanalyses used. We use observations from the
NASA's Earth Observing System (EOS) Aura satellite, launched on July 2004 into a polar sun synchronous orbit. In particular, we use OMI and MLS data. As mentioned before, we use the following meteorological reanalyses: MERRA, MERRA-2 ERA-Interim, JRA-55 and CFSR.

### 2.1  Aura-OMI

Aura-OMI (Levelt et al., 2006) is a nadir viewing push-broom spectrometer designed to monitor ozone and other trace gases,
as well as, aerosols, cloud top heights and UV irradiance at the surface. It measures ultraviolet/visible solar backscatter radiation with a high spectral resolution over the entire wavelength range from 270–500 nm. Total column ozone is derived using





two distinct algorithms, the Total Ozone Mapping Spectrometer (TOMS) algorithm and the differential optical absorption spectroscopy (DOAS) algorithm.

The OMI-TOMS algorithm is described by Bhartia and Wellemeyer (2002). This algorithm is based on the TOMS version 8 algorithm that has been used to estimate total ozone columns from four TOMS instruments since 1978. The OMI-DOAS

algorithm is described by Veefkind et al. (2006). Both datasets have been extensively validated (Balis et al., 2007; Kroon et al., 2008; McPeters et al., 2008; Antón et al., 2009) indicating agreement within 2% with ground-based and airborne measurements. In this study we use the OMI-TOMS algorithm. In particular, we use the level 3 files (OMTO3d, V003) with a 1° by 1° resolution.

## 2.2    Aura-MLS

Aura MLS (Waters et al., 2006) is a small radio-telescope whose mission objectives are studying ozone, air quality, and climate (Schoeberl et al., 2006). It vertically scans the Earth's limb from the surface to ∼95 km measuring thermal microwave emission with a spectral range varying from 118 to 2500 GHz. These radiances are inverted using a tomographic optimal estimation retrieval algorithm (Livesey et al., 2006), producing ∼3500 vertical profiles per day of temperature, cloud ice, and 16 atmospheric trace gases. In this study we use version 4.2 temperature and ozone data filtered as described in the MLS data

quality document (Livesey et al., 2017). This data-set provides ozone profiles from 261 to 0.02 hPa with vertical resolution of around 3 km in the upper troposphere and stratosphere, and precision varying from 0.03 ppmv at 261 hPa to 0.2 ppmv at 1 hPa and stratospheric accuracy better than 10%. Temperature is provided from 261 to 0.001 hPa with vertical resolution varying from around 4.5 km in the upper troposphere to 3.6 km in the middle stratosphere, and precision around 1 K at these levels and accuracy better than 2.5 K.

Ozone Version 2.2 was extensively validated (Jiang et al., 2007; Froidevaux et al., 2008; Livesey et al., 2008) indicating agreement at the 5 to 10% level with satellite, balloon, aircraft, and ground based ozone data. Above the tropopause, version 4.2 is very similar to version 2.2, so past validation results still hold. In the upper troposphere, version 4.2 has reduced spurious vertical oscillations found in previous versions, particularly at midlatitudes. Also, version 4.2 has reduced the ozone retrieval sensitivity to thick clouds through changes in the forward model representation of cloud impacts on the MLS radi-

ances (Livesey et al., 2017). Hubert et al. (2016) assessed the long term stability, overall bias, and short-term variability of several satellite ozone records using ground-based data and found MLS version 3.3 to be stable in the entire stratosphere (to within 1.5% decade$^{-1}$ in the middle stratosphere). Version 2.2 temperature data were extensively validated (Schwartz et al., 2008), indicating agreement at the 2.5 K level with satellite and radiosonde data, as well as with reanalysis fields. Version 4.2 is similar to version 2.2 so, again, the validation still holds.

## 2.3    Reanalyses

The reanalyses used in this study are MERRA and MERRA-2 (Rienecker et al., 2011; Bosilovich et al., 2015; Wargan et al., 2017), ERA-Interim (Dee et al., 2011), CFSR (Saha et al., 2010) and JRA-55 (Kobayashi et al., 2015). A detailed overview of these reanalyses is given by Fujiwara et al. (2017). Briefly, MERRA, MERRA-2 and CFSR use a 3D-FGAT ("first guess at the





appropriate time"; Lawless (2010)) assimilation scheme, while ERA-Interim and JRA-55 use an incremental 4D-Var (Courtier et al., 1994) approach. Overall, all reanalyses use the same conventional data (e.g., surface records, radiosonde profiles, and aircraft measurements); there are, however, many differences in the satellite data usage. In particular, the ozone inputs vary widely (see Figure 2): Only MERRA-2 and ERA-Interim assimilate OMI and MLS ozone data. Only MERRA-2 assimilates

MLS temperature retrievals at pressures less than or equal to 5 hPa. There are also differences in the horizontal and vertical grids, lid heights, and models' ozone treatment among the reanalyses. Table 1 summarizes these specifications.

To ease the comparison of the reanalysis fields against the OMI data, the reanalyses data were first interpolated to the OMI measurement times and then interpolated onto the OMI latitude-longitude grid, that is to say, a 1° latitude by a 1° longitude spacing. In addition, gaps in the OMI data (for example, polar winter periods) were identified and masked out in the interpolated

reanalysis fields to ensure that the same regions were compared day by day.

To ease comparison with MLS, we use the MLS DMPs (Manney et al., 2007). These DMPs are meteorological and derived meteorological fields interpolated to the MLS measurement locations (in time and space) computed within the JETPAC package (Manney et al., 2011), which also characterizes UTLS jets and multiple tropopauses. In particular, for this study we use ozone, temperature, equivalent latitude (EqL), and the JETPAC tropopause characterization. EqL is a quasi-Lagrangian coordinate

widely used in stratospheric studies (e.g., Butchart and Remsberg, 1986). Simply put, EqL is the latitude that would enclose the same area as the corresponding potential vorticity contours. Thermal tropopauses are determined from the reanalyses temperature profiles using the World Meteorological Organization (WMO) definition (e.g., Homeyer et al., 2010), that is to say, where the temperature lapse rate falls below $2 \, \mathrm{K \, km^{-1}}$ for at least 2 km. Similarly, additional tropopauses are identified above the primary tropopause (e.g., Randel et al., 2007; Añel et al., 2008; Manney et al., 2011).

## 3   Ozone mini-holes: Definition and Analysis

Several mini-hole definitions can be found in the literature: Hood et al. (2001) used a constant threshold of 215 DU, while Bojkov and Balis (2001) used 220 DU. James (1998a) used thresholds computed by subtracting 70 DU from zonally and meridionally averaged monthly means, while Iwao and Hirooka (2006) chose to subtract 80 DU. Koch et al. (2005) defined a threshhold based on monthly mean values minus one standard deviation, while Martínez-Lozano et al. (2011) used monthly

mean values minus two times the standard deviation. In addition to these thresholds, other constraints have been applied: Bojkov and Balis (2001) only considered as mini-holes those events with area greater than $500{,}000 \, \mathrm{km^2}$ in the $40^o$-$65^o$ latitude regions, while James (1998a) only considered events found over at least a $5.533^o$ latitude by $5.625^o$ longitude region, that is, covering an area equivalent to the ones found by Bojkov and Balis (2001).

Figure 3 shows the geographical distribution of mini-hole events found in the OMI data during 2005 using different mini-

hole definitions: Figure 3a shows the mini-hole geographical distribution found using a constant threshold of 220 DU; Figure 3b shows the events found using thresholds computed by subtracting 70 DU from the monthly mean; Figure 3c shows the events found using thresholds computed by subtracting two times the standard deviation from the monthly mean; Figure 3d shows the events found when the total column ozone value is less than 25% below the monthly mean. In each case, we use a





flood filling algorithm in the region of the total column ozone anomaly to find the adjacent pixels that were below the chosen threshold. Note than no additional constraints (size of the event or geographical position) were applied. Because of this, there are a disproportionate number of events in the southern hemisphere compared to the northern hemisphere in panels A, B, and D: Most of these events are related to the Antarctic ozone hole, that is to say, they are due to heterogeneous chemistry and not driven by dynamics; hence, we will not analyze them in this paper.

As shown in Figure 3, the occurrence frequency of mini-holes, as well as their geographical distribution, depends strongly on the definition used. In this study, we chose to define mini-hole events as regions where the total column ozone value is less than 25% below the monthly mean. Monthly means were used as opposed to climatological monthly means to avoid biasing the number of events by any long term trend in ozone or by interannual variability in the planetary scale "background" ozone values. We choose this definition because a constant — below 220 DU — threshold would identify more events in those months when the background ozone levels are naturally low. For example, OMI midlatitude mean total column ozone varied from ∼390 DU around March to ∼290 DU around mid-October. Hence, using a constant 220 DU threshold would identify more events during fall than in spring. An analogous argument applies to a fixed difference threshold, e.g., 70 DU less than the monthly mean. In a similar manner, the standard deviation threshold identifies many events in the tropics where ozone levels and variability are naturally low.

In addition to identifying the mini-holes in OMI and the reanalysis fields, the algorithm matches the events found in the reanalyses with the ones found in OMI. Within each day's events, the algorithm checks whether the events found in OMI and the reanalyses overlap; if they do not, the algorithm finds the closest one within a distance of 2000 km. Although this is a simple algorithm, visual inspection of many days showed it to be appropriate.

# 4    Comparison with OMI

Figure 4 compares the mini-hole events per month found in OMI and the reanalysis fields during 2005 to 2014. Figure 4a shows a timeseries of all the events found regardless of their area, while Figure 4b shows only events with area greater than 200,000 km$^2$. Clearly, the reanalyses are underestimating the frequency of the smaller events. Historically, the term mini-hole refers to synoptic-scale events, hence these sub-synoptic events have been referred as hindrance by James (1998a) and filtered out either by interpolating to synoptic-scale grids or by only analyzing events greater than a particular area (James, 1998a; Bojkov and Balis, 2001). In this study, we will only analyze events with area greater than 200,000 km$^2$ to avoid these micro-hole events. We note an increase in the number of micro-hole events after 2010; however, a detailed study of these events is beyond the scope of this paper.

Figure 4c shows that the mini-hole events' seasonal variations found in reanalyses and in the observations are similar. In OMI and the reanalysis fields, mini-hole events are most frequent during winter when the atmosphere is more dynamically active. Synoptic-scales storms are strongest and most common during midwinter, resulting in powerful storm tracks that uplift the air — increasing the amount of the column occupied by ozone-poor tropospheric air — which is one of the mechanisms responsible for mini-hole genesis (James, 1998a; Hood et al., 2001). Despite the similarities between the representation of mini-



holes in reanalyses and OMI data, differences exist among their seasonal variations: The most noticeable is that all reanalyses underestimate the number of mini-hole events, with the underestimation ranging from 34% less for ERA-Interim up to 83% less for JRA-55. Further, the events found in OMI display a mildly positively skewed distribution (the increase in number of events between September and January is rapid while the decay between January and March is slow), as opposed to the events found in the reanalyses (except for ERA-Interim), which display a distinctly negatively skewed distribution (the increase in number of events between September and March is slow, followed by a rapid decay in spring).

The geographical count of mini-hole events is shown in Figure 5. Although the reanalyses underestimate the number of mini-hole events, the mini-hole count morphologies are similar, with mini-holes occurring most frequently over the North Atlantic storm tracks. This region of maximum activity has been identified before by James (1998a) and Hood et al. (2001). This, in addition to the increase in mini-hole activity during winter, suggests that all reanalyses simulate the storm track influence upon mini-hole genesis to some degree.

Using the matching algorithm described above, it is possible to compute the distance between the matching events as well as to study their areas. Figure 6 compares the distance between the events found in the reanalysis fields and OMI data, as well as their area fractions. Ideally, one would like to have a delta function at zero when comparing the distance between events; the closest reanalysis to display this behavior is MERRA-2 which shows a narrower distribution centered around 75 km; the rest of the reanalyses display positively skewed distributions with the majority of values lying between 75 km and 300 km. With respect to their area fractions, ideally one would like to see a delta function at one; again, only MERRA-2 displays a narrow distribution, in this case near 0.8. Overall, the rest of the reanalyses usually underestimate the area of the mini-hole events. The slightly better performance in MERRA-2 may be related to the fact that this is not an independent comparison: MERRA-2 assimilates OMI total column ozone data throughout the comparison period. Note that ERA-Interim assimilates OMI data after 2008. However, CFSR only assimilates SBUV/2 ozone and performs similarly to ERA-Interim, suggesting that the dynamics produced by the reanalyses are more important than the assimilated ozone fields to reproducing mini-holes.

In addition to computing the distance between matching events and their area fraction, we also computed the direction that the events found in the reanalysis fields would have to move to match the position of the events found in the OMI data. Figure 7 summarizes the overall direction in which the mini-holes found in the reanalyses would have to move. The position of each pie slice indicates the direction in a polar coordinate system, its length represents the mean angular distance to be moved while its color represents the percentage of mini-hole matches in a particular direction. As can be seen, the mini-holes found in the CFSR, MERRA, MERRA-2, and ERA-Interim reanalyses display an eastward bias. That is, most of the time, these mini-holes would have to move westward to match the OMI event's positions.

## 5 Comparison with MLS

Comparisons with MLS allow us to study the vertical distribution of ozone and temperature during the events. Note that to increase the number of MLS co-locations with the mini-hole events we use day and night data. As a case study, Figure 8a shows the ozone vertical distribution for MLS during the mini-hole event shown in Figure 1. Figure 8a also shows a reference profile



constructed using profiles over the mini-hole region from 15 days prior to 15 days after the event excluding profiles under mini-hole events conditions. The ozone reduction occurs between 200 and 20 hPa. Following Keil et al. (2007), the profiles were split into two vertical regimes, an upper troposphere-lower stratosphere (UTLS) region (from 300 hPa to 65 hPa) and a mid-stratospheric (MS) region (from 65 hPa to 1 hPa). In each of these layers we computed the ozone decrease with respect to

the total column ozone. In the MLS data, about two thirds (67%) of the reduction occurs in the UTLS region while around one third (33%) originates in the mid-stratosphere. Using ozone sondes, Keil et al. (2007) found similar values (UTLS: 66%, MS: 34%).

In a similar manner, Figure 8b shows the vertical temperature distribution during the 19 January 2006 event. In this case, the mid-stratospheric temperatures and to some extent UTLS temperatures are lower than normal. Low temperatures in the

troposphere are usually associated with anticyclonic disturbances, which lead to local uplift of the air (Petzoldt et al., 1994). Using the tropopauses calculated by JETPAC, we computed the mean tropopause altitude during the event as well as during the reference period. The tropopause altitude found in MERRA-2 was 13.4 km±3.3 during the event as opposed to 11.3±1.6 km during the reference period, 2.1 km higher than normal. As indicated in section 1, raising of the tropopause leads to the replacement of relatively ozone-rich air in the column with tropospheric ozone-poor air, and, as pointed out by Petzoldt et al.

(1994), the uplift of air results in adiabatic cooling of the mid-stratosphere. For example, Hood et al. (2001) analyzed an ensemble of 71 extreme mini-holes (in this case, using a constant 215 DU threshold) and found a nearly linear relationship between the total column ozone and the 30 hPa temperature deviations.

Near the polar vortex edge, low temperatures in the mid-stratosphere are associated with planetary wave disturbances that are responsible for large-scale ozone redistribution (e.g., Leovy et al., 1985). As an example, Figure 8c shows the EqL vertical

profile derived using MERRA-2 potential vorticity fields. For this, we use the DMPs that, as mentioned before, have been interpolated to the MLS measurement locations. This EqL vertical distribution suggests that in the mid-stratosphere the air parcels originated at polar latitudes. To corroborate this, Figure 8d shows trajectories launched at 30 hPa from the MLS measurement locations during the mini-hole event. These trajectories were taken from the MLS Lagrangian Trajectory Diagnostic dataset (Livesey et al., 2015), which is a set of 15 day forward and 15 day reverse trajectories launched from a curtain of points along

the MLS track. These calculations are based on wind fields and diabatic heating rates taken from the MERRA-2 reanalysis and the advection calculations are based on the algorithm used by Manney et al. (1994). As expected, the majority of the air parcels in the mid-stratosphere originate near the polar vortex. Keil et al. (2007) computed back trajectories for the same event using the Met Office NAME III model and also found that the air in the mid-stratosphere was transported from the polar vortex, where it may have undergone ozone destruction (due to photo-chemical processes) before reaching the mini-hole event region.

Figure 8c, as well as the trajectories launched at 100 hPa shown in panel d, indicates that the air parcels in the UTLS originated at low latitudes. Studies of the characteristics of poleward advection of upper tropospheric air have shown that such intrusions are associated with Rossby wave breaking in the upper troposphere (Peters and Waugh, 1996). In turn, poleward Rossby wave breaking has been associated with the presence of double tropopauses (DT) (Pan et al., 2009; Castanheira and Gimeno, 2011; Homeyer et al., 2011; Ungermann et al., 2013). Further, climatological studies have found that DT occurrence,

in the northern hemisphere coincides with zones of storm track cyclogenesis (e.g., Añel et al., 2008; Peevey et al., 2012)



and their occurrence frequency shows a strong seasonal variation peaking during winter (Randel et al., 2007; Manney et al., 2014); both characteristics displayed by the mini-hole events. Using the JETPAC tropopause information, we computed the DT fraction during the event as well as during the reference period. During the event, the DT fraction found in MERRA-2 was $0.8\pm0.4$, as opposed to $0.61\pm0.49$ during the reference period.

Figure 9 displays a composite view of ozone, temperature and EqL vertical distributions, as well as their total ozone column, tropopause altitude, and DT deviations, for all the events found between 2005 and 2014 in the northern hemisphere. The number of events is shown in Figure 6. For ozone and temperature, two composites are shown, one smoothed with the MLS averaging kernels and one without; note that no significant differences were found between the two. The observations and reanalysis fields show a picture that is generally consistent with the one shown in Figure 8:

– The total ozone column decrease is considerably larger than the natural variability of total ozone column, that is, it is considerably larger than the reference standard deviation of total ozone column.

    – On average, around two thirds of the reduction originates in the UTLS and the rest in the mid-stratosphere.

    – Air parcels in the UT originate from low latitudes, while in the mid-stratosphere they arrive from high latitudes.

    – Reanalyses show an elevated tropopause during the events. This is consistent with anticyclonic disturbances associated
with poleward Rossby wave breaking in the upper troposphere, that is, poleward advection of upper tropospheric air.

    – The local uplift of air adiabatically cools the MS, resulting in lower than normal 30 hPa temperatures.

    – An increase in DT fraction is seen during the events. However, this increase is considerably smaller than the DT natural variability. This may be because even though DT may be dynamically coupled with Rossby wave breaking events, DTs occur most frequently above strong cyclonic circulation system (Randel et al., 2007). That is, mini-hole events are
associated with anticyclonic Rossby wave breaking and thus, while some DTs are favorable for mini-hole genesis, many others are not.

As shown in Figure 9, ozone, temperature, and EqL vertical distributions, as well as the tropopause altitude, during the events are close to, or sometimes inside, the limits of their natural variability. This suggests that mini-hole events are only produced when both the UTLS and mid-statosphere processes are favorable for reduction of ozone by dynamical processes.

**6  Summary**

Dynamical redistribution of ozone can produce large transient and localized ozone reductions, also known as mini-holes. In this study we analyze the representation of mini-hole events in the northern hemisphere from several reanalyses (ERA-Interim, MERRA, MERRA-2, CFSR and JRA-55) using data from OMI and MLS. OMI data allow us to compare their geographical representation while MLS data allow us to study their vertical representation.

Several definitions of mini-holes exist in the literature. The results presented here show that the mini-hole frequency as well as their geographical distribution differs vastly depending on their definition. Here, we define mini-hole events as regions





where the total column ozone value is less than 25% below the monthly mean. Further, we only consider as mini-hole events those ozone fluctuations with an area larger than 200,000 km$^2$.

The main findings can be summarized as follows:

– OMI and the reanalysis fields display the same mini-hole seasonal variability, with more mini-hole events during winter when the atmosphere is more dynamically active.

– OMI and the reanalysis fields display similar mini-hole geographical distributions with mini-holes occurring more frequently over the North Atlantic storm tracks.

– All reanalyses underestimate the number of mini-hole events, with the underestimation ranging from 34% less for ERA-Interim up to 83% less for JRA-55. Further, reanalyses typically underestimate the area of the mini-hole events and most of the time are between 75 km and 300 km away from the events found in OMI.

– Mini-holes found in CFSR, MERRA, MERRA-2 and ERA-Interim reanalyses display an eastward bias with respect to the events found in OMI data.

– The composite view of the vertical representation of the events agrees with previously reported mechanisms for dynamical mini-hole formation: Anticyclonic poleward Rossby wave activity breaking into the UTLS and local uplift of air brings ozone poor air into the column and is accompanied by equatorward advection of polar air in the mid-stratosphere.

– On average, in the events found in both MLS and the reanalyses, around two thirds of the ozone reduction originates in the UTLS and the rest in the mid-stratosphere.

– Although mini-hole regions typically show more DTs than in surrounding air, the association is not strong because because DTs occur most frequently above strong cyclonic circulation system while mini-holes occur above anticyclonic systems.

In general, MERRA-2 seems to represent mini-holes more accurately than the rest of the reanalyses, likely because MERRA-2 assimilates OMI and MLS ozone throughout the comparison period. Independent comparisons performed by Wargan et al. (2017) suggest that MERRA-2 upper tropospheric and stratospheric ozone are of sufficient quality for studies requiring high frequency, highly resolved global ozone maps with variability consistent with dynamics. CFSR assimilates only SBUV/2 ozone, and performs similarly well to ERA-Interim, which assimilates OMI and MLS ozone during 2008 and after mid-2009. This suggests that the dynamics produced by the reanalyses are more important than the assimilated ozone fields in reproducing mini-holes.

## 7   Data availability

All the data and reanalysis fields used in this study are publicly available. Reanalysis fields can be found at NASA GMAO, ECMWF, JMA and NCEP websites. MLS and OMI data are available from the NASA Goddard Space Flight Center Earth Sciences (GES) Data and Information Services Center (http://disc.sci.gsfc.nasa.gov/Aura/data-holdings/).





*Acknowledgements.* We thank the JPL MLS team, especially B. Knosp and R. Fuller, as well as Z. Lawrence, for help in obtaining, managing, and processing the reanalysis datasets; N. Livesey for supplying the MLS Lagrangian Trajectory Diagnostic dataset; NASA's GMAO, ECMWF, JMA, and NCEP for providing their reanalysis data; and M. Hegglin for helpful discussions. Work at the Jet Propulsion Laboratory, California Institute of Technology, was done under contract with the National Aeronautics and Space Administration.





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





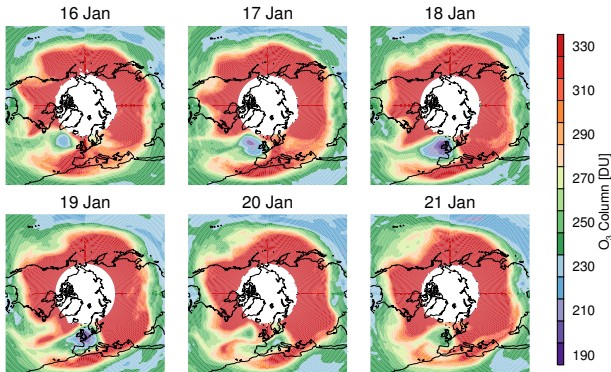

**Figure 1.** OMI total ozone column for the 16 to 21 January 2006. The missing data over the pole is due to the lack of UV backscattering. Red/Purple indicate relatively high/low values of OMI total ozone column.

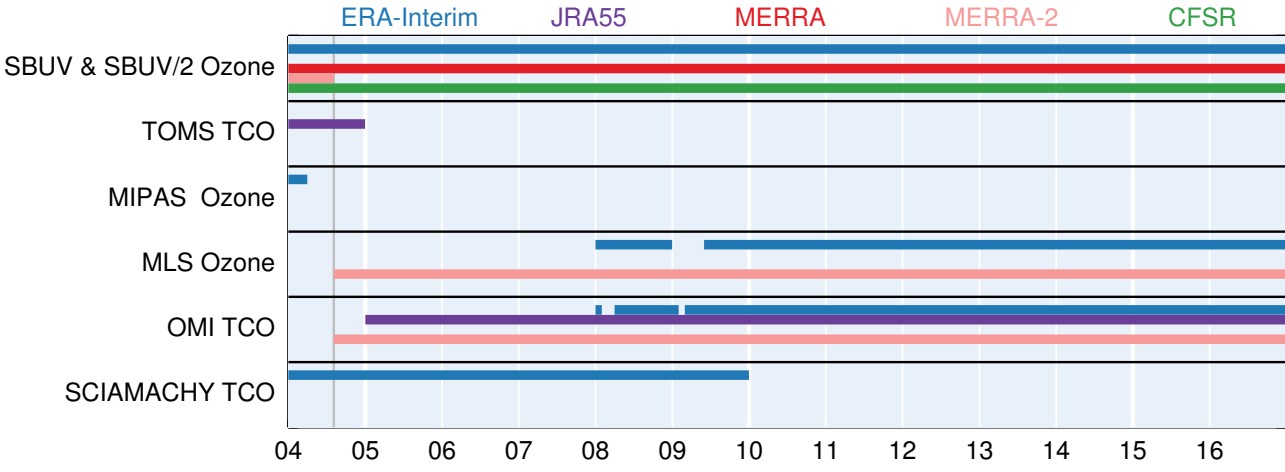

**Figure 2.** Timelines of ozone observations (vertical profiles and total column ozone – TCO) assimilated during the MLS/OMI period by the ERA-Interim (blue), JRA55 (purple), MERRA (dark-red), MERRA-2 (light-red), and CFSR (green) reanalysis systems. Based on Figure 9 from Fujiwara et al. (2017). Gray vertical line indicates the start of the OMI and MLS measurements.





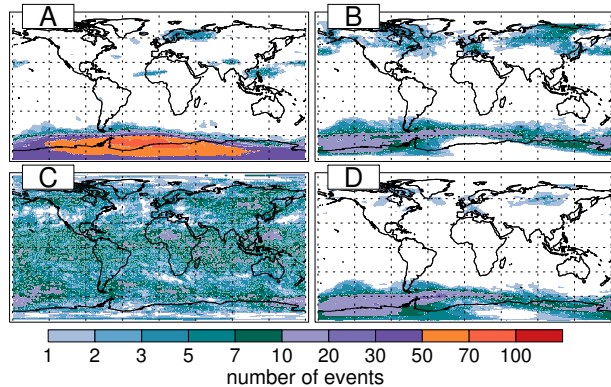

**Figure 3.** Geographical distribution of mini-hole events for OMI data in 2005. Four different mini-hole definitions are compared; A) using a constant threshold of 220 DU, B) using as thresholds the monthly mean minus 70 DU, C) using as thresholds the monthly mean minus two times its standard deviation and, D) the percentage threshold discussed in the text. Red/blue indicate relatively high/low number of events counts.

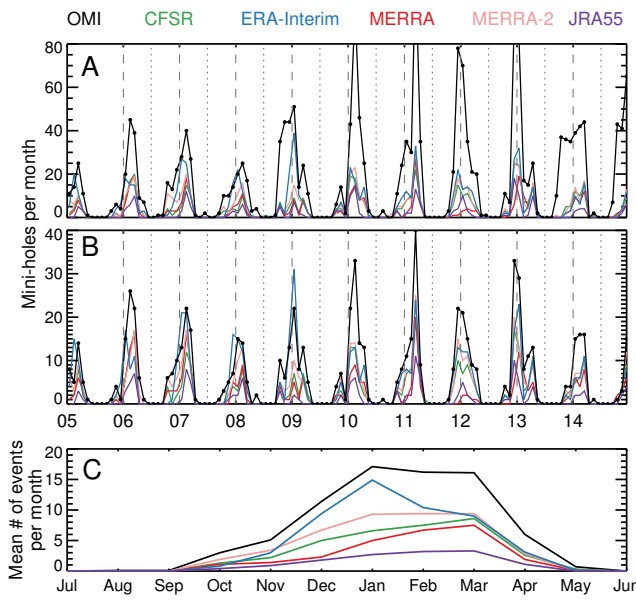

**Figure 4.** Mini-hole events per month in the northern hemisphere as found in OMI data and reanalysis fields (Black, green, blue, red, pink, purple lines represent OMI, CFSR, ERA-Interim, MERRA, MERRA-2, and JRA-55 respectively). Panel A shows all the events while panel B displays only the events with area greater than 200,000 $km^2$. Dashed vertical lines indicate the beginning of each January, dotted vertical lines the beginning of each July. Panel C shows the mean number of mini-hole events in a given month for events with area greater than 200,000 $km^2$.




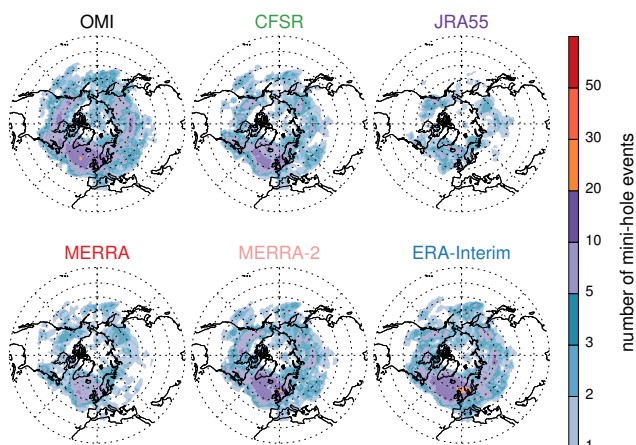

**Figure 5.** Geographical distribution of mini-hole events during 2005-2014 as found in OMI and reanalysis fields. Red/blue indicate relatively high/low number of events counts.

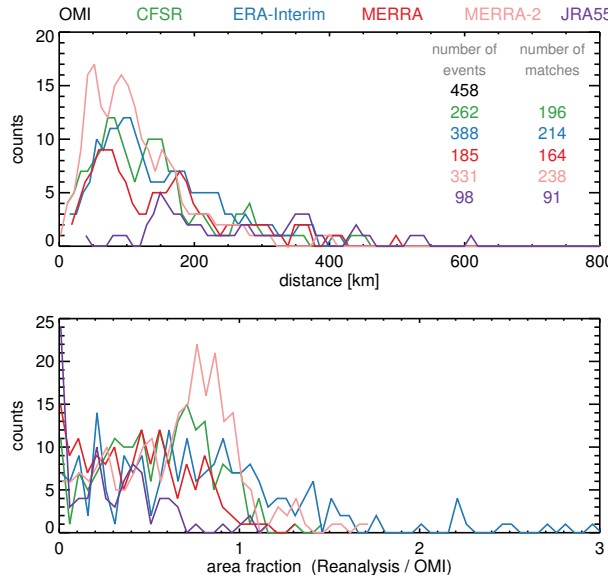

**Figure 6.** (Top) Histograms of the distance between the mini-hole events found in the reanalysis fields and the ones found in OMI data (Black, green, blue, red, pink, purple lines represent OMI, CFSR, ERA-Interim, MERRA, MERRA-2, and JRA-55 respectively). Also shown is the total number of events as well as the number of matches between the events found in OMI and in the reanalyses. (Bottom) Histograms of the area fraction of mini-hole events.





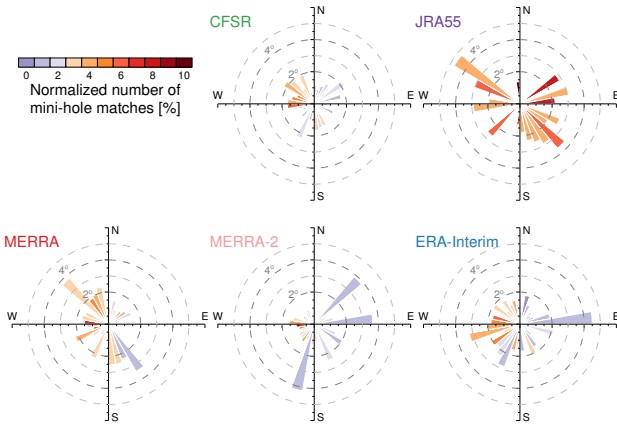

**Figure 7.** Wind rose plots showing the direction in which events found in the reanalyis fields would have to move to match the events found in OMI data, as well as the mean angular distance to be moved in a particular direction. Gray dashed circles show angular distance. Red/blue indicate relatively high/low normalized number of matches.

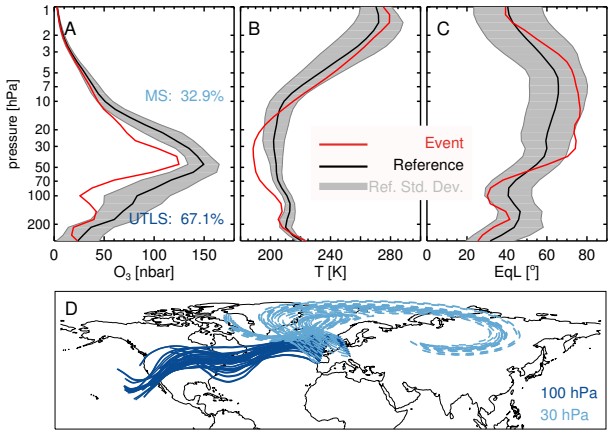

**Figure 8.** A) MLS mean ozone vertical profile (red), reference ozone profile (black) and reference standard deviation (gray envelope) during the mini-hole event observed over the UK on the 19 of January 2006. The percentage ozone reduction with respect to the total column ozone in the UTLS and in the mid-stratosphere is shown in dark and light blue respectively. B) As in panel A but for Temperature. C) As in panel A but for equivalent latitude (EqL) derived from the MERRA-2 reanalysis sampled at the MLS measurement locations. D) MERRA-2 trajectories launched at the MLS measurement locations in the mini-hole event region at 100 hPa and at 30 hPa.



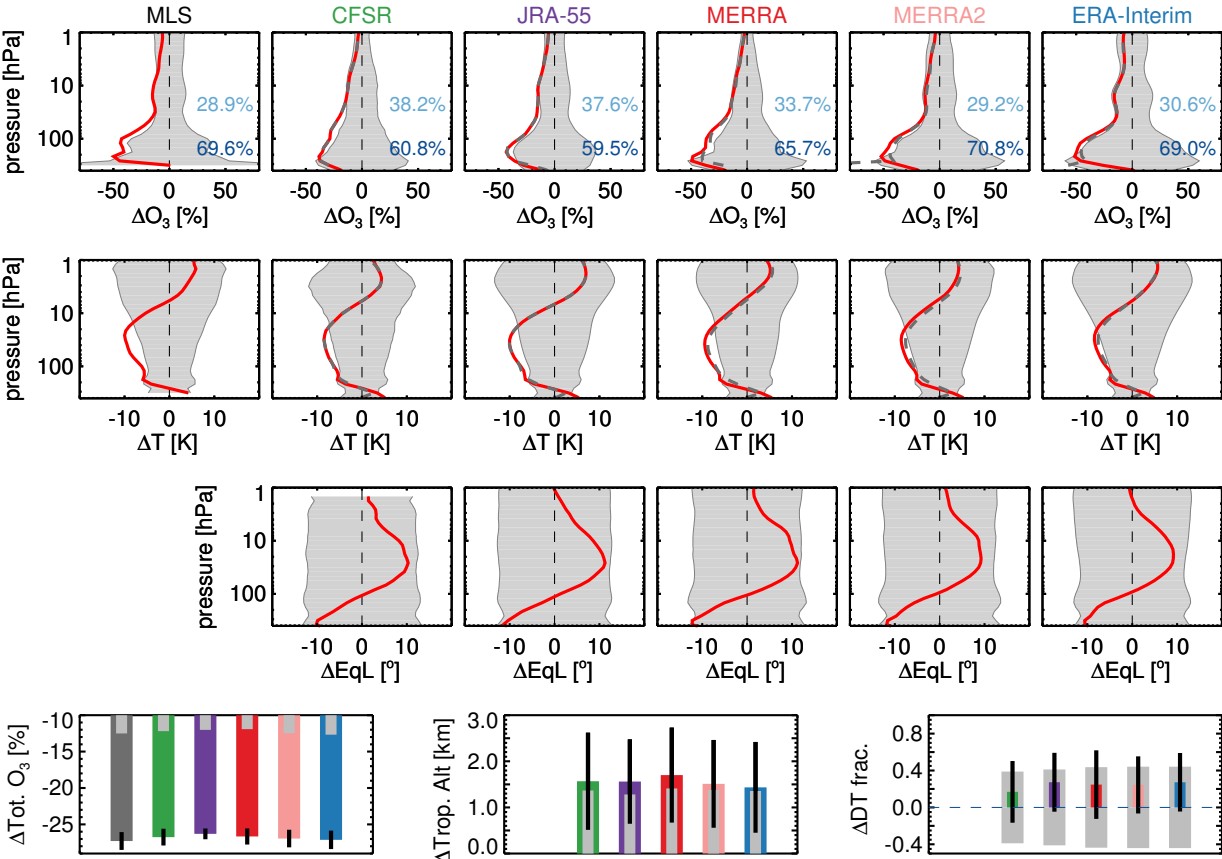

**Figure 9.** Composite of the difference between the events and the reference values for all mini-hole events found between 2005 and 2014. The first row shows the ozone vertical profile differences. The ozone composite event difference profile is shown in red, the gray envelope is the reference composite standard deviation. In the reanalyses panels, the blue dashed lines show the ozone composite event difference when the MLS averaging kernels were applied. The percentage ozone reduction with respect to the total column of ozone in the UTLS and in the mid-stratosphere are shown in dark and light blue respectively. Similarly, the second row displays the temperature profile differences and the third row displays the EqL vertical profile differences. Lastly, the fourth row shows the percentage difference of the total ozone column, the tropopause altitude difference, and the DTs fraction deviations (Dark gray, green, blue, red, pink, purple bars represent the OMI, CFSR, ERA-Interim, MERRA, MERRA-2, and JRA-55 differences respectively). In this column, light gray bars show the composite reference standard deviation, color bars show the corresponding composite event difference, and the black lines display the composite difference event standard deviation.





**Table 1.** Basic specifications of the reanalysis forecast models.

| Reanalyses | Grid | # levels | Lid Height | Reference | Ozone model |
|---|---|---|---|---|---|
| MERRA | 0.66°x0.5° | 72 | 0.01 hPa | Rienecker et al. (2011) | Rienecker et al. (2008) |
| MERRA-2 | 0.625°x0.5° | 72 | 0.01 hPa | Bosilovich et al. (2015) | Rienecker et al. (2008) |
| ERA-Interim | 0.75°x0.75° | 60 | 0.1 hPa | Dee et al. (2011) | Cariolle and Déqué (1986); Dethof and Hólm (2004); Cariolle and Teyssèdre (2007) |
| CFSR | 0.5°x0.5° | 64 | ∼0.26 hPa | Saha et al. (2010) | McCormack et al. (2006) |
| JRA-55 | 0.56°x0.56° | 60 | 0.1 hPa | Kobayashi et al. (2015) | Shibata et al. (2010) |