# Peer review of "An assessment of Ozone Mini-holes Representation in Reanalyses Over the Northern Hemisphere"

_Atmospheric Chemistry and Physics, 2017_

## Referee Comment (RC1) · Anonymous Referee #1 · 2 May 2017

This is a quality study of the ability of meteorological reanalyses to simulate ozone-miniholes. Most of the conclusions are well supported by the presented results. In particular, it is inferred that the dynamics of the reanalysis models at synoptic scales is deficient and that this is the main reason why the reanalyses do a poor job of simulating ozone mini-holes. My only real negative comment concerns the last paragraph of the Conclusions section on page 9. It states that: "In general, MERRA-2 seems to represent mini-holes more accurately than the rest of the reanalyses." I don't understand where this statement is supported in the rest of the text. For example, it is noted several times that ERA-Interim underestimates the number of mini-hole events the least among the reanalyses (34 per cent). Figure 4 shows that the number of events per month is

underestimated less by ERA-Interim than by MERRA-2. Only in Figure 6 is there any indication that MERRA-2 might do slightly better in identifying specific mini-holes found in the OMI measurements (238 matches as compared to 214 for ERA-Interim). So, I think this last paragraph needs to be modified to at least say that ERA-Interim does about as well as MERRA-2. ERA-Interim also assimilates the OMI and MLS ozone data.

Overall, the final paragraph of the conclusions section and the data availability section 7 sound a little too much like an advertisement for using reanalysis data to study ozone variability on short time scales. It mentions that "Independent comparisons performed by Wargan et al. (2017) suggest that MERRA-2 upper tropospheric and stratospheric ozone are of sufficient quality for studies requiring high-frequency, highly resolved global ozone maps and variability consistent with dynamics." This sentence needs to be removed in this reviewer's opinion. The results of this study clearly show that none of the reanalyses is of sufficient quality for this purpose. Any future studies of ozone mini-holes, for example, must use actual satellite ozone measurements such as OMI. This should be the main conclusion of the paper. Please revise the text.

---

## Author Comment (AC1) · 9 May 2017

Luis Millan and Gloria Manney

lmillan@jpl.nasa.gov

We thank the reviewer for his/her comments. Below is our response.

Reviewer comments:

This is a quality study of the ability of meteorological reanalyses to simulate ozone miniholes. Most of the conclusions are well supported by the presented results. In particular, it is inferred that the dynamics of the reanalysis models at synoptic scales is deficient and that this is the main reason why the reanalyses do a poor job of simulating ozone mini-holes. My only real negative comment concerns the last paragraph of the Conclusions section on page 9. It states that: "In general, MERRA-2 seems

to represent mini-holes more accurately than the rest of the reanalyses." I don't understand where this statement is supported in the rest of the text. For example, it is noted several times that ERA-Interim underestimates the number of mini-hole events the least among the reanalyses (34 per cent). Figure 4 shows that the number of events per month is underestimated less by ERA-Interim than by MERRA-2. Only in Figure 6 is there any indication that MERRA-2 might do slightly better in identifying specific mini-holes found in the OMI measurements (238 matches as compared to 214 for ERA-Interim). So, I think this last paragraph needs to be modified to at least say that ERA-Interim does about as well as MERRA-2. ERA-Interim also assimilates the OMI and MLS ozone data. Overall, the final paragraph of the conclusions section and the data availability section 7 sound a little too much like an advertisement for using reanalysis data to study ozone variability on short time scales. It mentions that "Independent comparisons performed by Wargan et al. (2017) suggest that MERRA-2 upper tropospheric and stratospheric ozone are of sufficient quality for studies requiring high-frequency, highly resolved global ozone maps and variability consistent with dynamics." This sentence needs to be removed in this reviewer's opinion. The results of this study clearly show that none of the reanalyses is of sufficient quality for this purpose. Any future studies of ozone mini-holes, for example, must use actual satellite ozone measurements such as OMI. This should be the main conclusion of the paper. Please revise the text.

Response:

The last paragraph in the conclusion will be modified to:

In general, MERRA-2 seems to represent mini-holes marginally better than the rest of the reanalyses (see Figure 6 and 7), likely because MERRA-2 assimilates OMI and MLS ozone throughout the comparison period. CFSR assimilates only SBUV/2 ozone, and performs similarly well to ERA-Interim, which assimilates OMI and MLS ozone during 2008 and after mid-2009. This suggests that the dynamics produced by the reanalyses are more important than the assimilated ozone fields in reproducing miniholes.

Because of the mismatch between the mini-holes found in OMI and the mini-holes found in the reanalysis fields, careful attention needs to be paid to ensure that the regions used to study them coincide with regions where the reanalysis fields display mini-hole conditions. That is to say, it is insufficient to identify the mini-hole position in the data and then see what the reanalysis fields do at those exact locations. Rather, one should find the nearby mini-holes in the reanalysis fields, see whether their characteristics (magnitude, timing) are similar to those in the OMI data, and if they are, study the meteorological conditions at those locations. Because none of the reanalyses fully captures mini-hole features and distributions, the satellite data remain an essential tool for studying ozone mini-holes.

---

## Referee Comment (RC2) · Anonymous Referee #2 · 22 May 2017

The paper by Millan and Manney deals with the representation of so-called 'ozone mini-holes' within several widely used reanalyses data sets. The topic of ozone mini-holes and their dynamics is relevant for ACP, as well as an investigation of the respective capabilities of the reanalyses. The manuscript is well written and presents the findings in a clear manner. The manuscript can be published to ACP after taking into account the below-mentioned comments.

Specific comments:

P1 L24, P5, L31, 32, P7 L10-15: 'Local uplift of the air . . .' Indeed, this is an important process for generating ozone mini-holes. However, the total ozone column amount is only decreased if there is also net-divergence of air out of the column which compensates for the air parcel expansion. Otherwise, the ozone is just vertically re-distributed. This could be checked by investigating if indeed the pressure difference between isentropes decreases. This should normally be the case during ozone mini-hole events, but could the authors perform a check on this?

P2, last paragraph of introduction: The authors should mention why it is relevant to investigate the capabilities of reanalyses to reproduce ozone min-holes, where is the advantage for the scientific community to characterize this?

P3 L22, L29: please give a source/reference for the statement that MLS version 4.2 is very similar to version 2.2. It also needs only once be stated.

P5 L26, 27: 'micro-hole events'. I suggest to avoid to introduce another naming besides 'mini-holes' because there are already several definitions for 'mini-holes'. What would then be a 'micro-hole'? Better write '...smaller events...' or '... smaller-scale events ...'

P6 L7-11: discussion of Figure 5: I think that with a difference-plot (OMI minus reanalysis model) more information could be gained and the last sentence of this paragraph could be more precise than now with 'suggests', 'to some degree'.

P6 L19-22: Here the authors could go more into detail. E.g., JRA55 assimilates OMI also nearly from the beginning, as MERRA-2, but performs not that good. Are there other studies which investigate the general ozone field performance of the reanalyses? Further, in Figure 6 the number of detected events and matches are given – could the authors give some information/discussion why there are so clear differences between the reanalyses? The relative score of matches could also be discussed, e.g., JRA55 detects the least OMI events, but has the relatively highest score of matches (out of its detected ones).

P6 L24: '... move to match the position of the events...': When is this 'match' reached exactly? Do the authors define a centre of the OMI event?

[Figure]

P6 L27-29: Here, the authors could give more details. The shift/move would often have to be to NW, SW, not only pure East. Can this be linked to general atmospheric dynamics/circulation? And, JRA55 shows a different behavior. What could be reasons for these patterns?

P8 L3: 'DT fraction': what is the reference?, fraction of what? Please clarify.

P8 L6: '. . . for all the events. . .': Please clarify if indeed events or only matches are meant.

P8 L10, 1/figure 91: would it be possible to show (or to mention) also the error of the (red) ozone, temperature, EqL composite event difference profile?

P8 L12/Figure 9: fraction of reduction origin: from the percentages given, MERRA-2 and ERA-Interim agree much better with MLS than the other three reanalyses. Could you please comment on this?

P8 L23, 24: 'This suggests . . .'. It would be helpful to support this further. E.g., the authors could calculate the MS part of reduction and see if there would still be a mini-hole following their definition of ozone mini-holes. Respective findings could also be used to rephrase in the summary p9 L26.

P9 L11, 12: please mention also JRA55 here

P9 L16, 17: Concerning the average, this statement is correct. But for the reader it would me more helpful to mention that there are nonetheless differences between the five reanalyses concerning that distribution.

Technical comments:

P2 L7: 'normally is around . . .' please replace with 'the long-term mean around 310 DU during this time . . .'

P2 L10: ' . . . based on an . . .'

[Figure]

P2 L17: please refer to table 1 after the listing of the five reanalyses P2 L27: dito

P6 L15: please change '. . . the rest' to '. . . the other reanalyses. . .'

P8 L19 and P9 L19: ' . . . strong cyclonic circulation systems . . .'

P9 L18/19: please delete one 'because'

———————————————

---

## Author Comment (AC2)

Respond to referee 2:
We thank the reviewer for his/her comments. Below is our response in blue:

Reviewer comments:

The paper by Millan and Manney deals with the representation of so-called 'ozone miniholes' within several widely used reanalyses data sets. The topic of ozone mini-holes and their dynamics is relevant for ACP, as well as an investigation of the respective capabilities of the reanalyses. The manuscript is well written and presents the findings in a clear manner. The manuscript can be published to ACP after taking into account the below-mentioned comments.

Specific comments:
P1 L24, P5, L31, 32, P7 L10-15: 'Local uplift of the air : : :' Indeed, this is an important process for generating ozone mini-holes. However, the total ozone column amount is only decreased if there is also net-divergence of air out of the column which compensates for the air parcel expansion. Otherwise, the ozone is just vertically re-distributed.    This could be checked by investigating if indeed the pressure difference between isentropes decreases. This should normally be the case during ozone mini-hole events, but could the authors perform a check on this?
We performed the requested check:   in MLS as well as in the reanalyses there is a decrease in pressure (against the reference period) between the 330 and 500K isentropes varying from -6.5hPa for MERRA2, -8.3 for MLS and up to – 8.9hPa in JRA-55.

P1L25 will be changed to (changes in bold):  **Assuming net divergence**, local uplift of air …

P5L31 will be changed to (changes in bold):  --- which**, assuming net divergence,** increases the amount of the column occupied by ozone poor tropospheric air ---

P7L10 will be changed to (changes in bold): As indicated in section 1, **assuming divergence of air** raising of the tropopause leads the replacement of relatively ozone-rich air in the column with tropospheric ozone poor air. **To verify that there was net divergence, we estimated the pressure between the isentropes 330K and 500K; we found 168.4 hPa during the event versus 170.9hPa during the reference period.**   As pointed out by Petzold et al. (1994) ….

In P8L15 we will add: Note that, overall, net divergence of air was present during the events, with a pressure difference between the 330 and 500K isentropes varying from 6.5hPa to 8.9 hPa less than that in the reference period depending on the data source analyzed.

P2, last paragraph of introduction: The authors should mention why it is relevant to investigate the capabilities of reanalyses to reproduce ozone min-holes, where is the advantage for the scientific community to characterize this?
We will add the following paragraph at the end of the introduction: Dynamically induced ozone mini-holes can produce extreme ozone deficits that result in significant local increases in surface UV; thus our ability to predict and characterize these events is important to human health.  Because of the combination of dynamical and transport processes that produce mini-holes, they are a stringent test of the representation of UTLS dynamics in the reanalyses.  Understanding of the reanalyses' ability to reproduce these events can thus be used to guide improvements in the models and data assimilation systems, and hence in our ability to forecast such events.

P3 L22, L29: please give a source/reference for the statement that MLS version 4.2 is very similar to version 2.2. It also needs only once be stated. Livesey et al. (2017) will be added at the end of the paragraph.

P5 L26, 27: 'micro-hole events'. I suggest to avoid to introduce another naming besides 'mini-holes' because there are already several definitions for 'mini-holes'. What would then be a 'micro-hole'? Better write ':::smaller events: : :' or ': : : smaller-scale events : : :' Micro-hole event will be changed to sub-synpotic scale events.

P6 L7-11: discussion of Figure 5: I think that with a difference-plot (OMI minus reanalysis model) more information could be gained and the last sentence of this paragraph could be more precise than now with 'suggests', 'to some degree'. The figure suggested by the reviewer can be found below. We believe that not much information is gained by using it, the underestimation is clearly seen in this figure, however, it is also showcase in the original figure and in Figure 4 in the paper. Further, in this new figure it is harder to see that all reanalysis have similar count morphologies with maxima over the north atlantic storm tracks. Hence, after careful consideration, we decided to leave Figure 5 as is.

[Figure]

As in Figure 5 of the paper but showing the differences (Reanalysis – OMI).

P6 L19-22: Here the authors could go more into detail. E.g., JRA55 assimilates OMI also nearly from the beginning, as MERRA-2, but performs not that good. In reality JRA-55 does not assimilate directly the OMI measurements. The following sentence will be added in P4L6: Note that JRA-55 does not assimilate measurements directly, first, ozone concetrations are estimated using a chemistry transport model and then nudged to the total colum ozone observations (Kobayashi et al. 2015, Fujiwara et al. 2017).

In addition, a similar sentence will be added in the caption of Figure 2: Note that JRA-55 does not assimilate OMI TCO directly, first, ozone concetrations are estimated using a chemistry transport model and then nudged to the TCO observations (Kobayashi et al. 2015, Fujiwara et al. 2017).

Reference: Kobayashi et al: The JRA-55 Reanalysis: General Specifications and Basic Characteristics, J Meteorol Soc, doi:10.2151/jmsj.2015-001, 2015.

Are there other studies which investigate the general ozone field performance of the reanalyses?

The following sentence will be added in P4L6:   An assessment of the upper tropospheric and stratospheric reanalyses ozone fields can be found at Davis et al. 2017.

Reference: Davis et al: Assessment of upper tropospheric and stratospheric water vapour and ozone in reanalyses as part of S-RIP, Atmos. Chem. Phys. Discuss.,  10.5194/acp-2017-377, in review, 2017.

Further, in Figure 6 the number of detected events and matches are given – could the authors give some information/discussion why there are so clear differences between the reanalyses?  The relative score of matches could also be discussed, e.g., JRA55 detects the least OMI events, but has the relatively highest score of matches (out of its detected ones).   The relative scores will be added to figure 6 (see below) and the following paragraph will be added: Figure 6 also shows the number of events found in each dataset, the number of matching events as well as their relative score (number of matches divided by their total number of events).  Despite having a similar number of matches, MERRA2, CFSR and ERA-Interim have different relative scores, 0.72, 0.75, and 0.55, respectively. ERA-Interim's low relative score indicates that half of its minihole events were not found in OMI in contrast to around a third of those in MERRA2 or CFSR. This indicates that although ERA-Interim underestimates the number of events least, many of the events in ERA-Interim, are not found in the OMI data. Note that, the reltatively high scores of JRA55 and MERRA result from those reanalyses detecting mostly the strongest minihole events that are in OMI, whereas the low number of events indicate that those reanalyses do poorly at detecting the smaller events seen in OMI data.

[Figure]

P6 L24: ': : : move to match the position of the events: : :': When is this 'match' reached exactly? Do the authors define a centre of the OMI event? Yes, we did, the sentence will be changed to: In addition to computing the distance between matching events and their area fraction, we also computed the direction that the events found in the reanalysis fields would have to move to match the position (the latitude-longitude center) of the events found in the OMI data.

P6 L27-29: Here, the authors could give more details. The shift/move would often have to be to NW, SW, not only pure East. Thats why the senteces talk in terms of eastward bias or move westward rather than east bias or move west.

Can this be linked to general atmospheric dynamics/circulation? Yes, one could speculate that this might point out to the reanalyses having too strong westerlies which shift the reanalysis events towards the east. We will add the following at the end of that section. This suggests the possibility that the reanalyses could have westerlies that are too strong, which would shift the reanalyses events eastward however investigating this in detail would require extensive study that is beyond the scope of this paper.

And, JRA55 shows a different behavior. What could be reasons for these patterns? The following sentences will be added at the end of the paragrah: JRA55 does not show a particular bias direction, that is, individual JRA55 miniholes have to move in different directions to match the OMI events' positions, this is more likely related to their crude threatment of ozone.

P8 L3: 'DT fraction': what is the reference?, fraction of what? Please clarify. We will add (changes in bold): Using the JETPAC tropopause information, we computed the DT fraction **(the area with double tropopauses divided by the total area)** during the event as well as during the reference period.

P8 L6: ': : : for all the events: : :': Please clarify if indeed events or only matches are meant. We do mean events, the sentence will be changed to (changes in bold): … for all the events, **that is not only the matches**, found between 2005...

P8 L10, 1/figure 91: would it be possible to show (or to mention) also the error of the (red) ozone, temperature, EqL composite event difference profile? Below is the new figure, which includes the error. The following sentence in the caption will be changed to (changes in bold): The ozone composite event difference profile is shown in red, **its one standard deviation envelope is shown by the thin light red lines**, and the gray envelope is the reference composite standard deviation.

[Figure]

P8 L12/Figure 9: fraction of reduction origin: from the percentages given, MERRA-2 and ERA-Interim agree much better with MLS than the other three reanalyses. Could you please comment on this? This is probably because MERRA2 assimilates MLS throughout out the comparison period and ERA-Interim from 2008 (excepting the first half of 2009). The following sentences will be added: MERRA2 and ERA-Interim show very close agreement with the MLS estimates, presumably because both assimilate, in some capacity, the MLS O3 profiles; however, the other reanalysis also show good agreement.

P8 L23, 24: 'This suggests : : :'. It would be helpful to support this further. E.g., the authors could calculate the MS part of reduction and see if there would still be a minihole following their definition of ozone mini-holes. Respective findings could also be used to rephrase in the summary p9 L26. As suggested by the reviewer, we estimated the reduction due to UTLS and MS only for all the events. Neither in the MLS data nor in the reanalysis fields was the UTLS nor the MS reduction enough to produce a minihole event. We will add the following at the end of the paragraph (P8L24): To verify this, we investigated if the UTLS or the MS part of the reduction was enough to produce a minihole event, that is a 25% reduction below the monthly mean. We found that, neither in the MLS data nor in the reanalysis fields was the UTLS or the MS reduction enough to produce a single minihole event.

The summary (P9L26) will be changed as follows in response to a suggestion from reviewer 1: In general, MERRA-2 seems to represent mini-holes marginally better than the rest of the reanalyses (see Figure 6 and 7), likely because MERRA-2 assimilates OMI and MLS ozone throughout the comparison period. CFSR assimilates only SBUV/2 ozone, and performs similarly well to ERA-Interim, which assimilates OMI and MLS ozone during 2008 and after mid-2009. This suggests that the dynamics produced by the reanalyses are more important than the assimilated ozone fields in reproducing miniholes.

Because of the mismatch between the mini-holes found in OMI and the mini-holes found in the reanalysis fields, careful attention needs to be paid to ensure that the regions used to study them coincide with regions where the reanalysis fields display mini-hole conditions. That is to say, it is insufficient to identify the mini-hole position in the data and then see what the reanalysis fields do at those exact locations. Rather, one should find the nearby mini-holes in the reanalysis fields, see whether their characteristics (magnitude, timing) are similar to those in the OMI data, and if they are, study the meteorological conditions at those locations. Because none of the reanalyses fully captures mini-hole features and distributions, the satellite data remain an essential tool for studying ozone mini-holes.

P9 L11, 12: please mention also JRA55 here. The following sentence will be added: JRA55 does not show a clear bias direction most likely related to their crude treatment of ozone.

P9 L16, 17: Concerning the average, this statement is correct. But for the reader it would me more helpful to mention that there are nonetheless differences between the five reanalyses concerning that distribution. Since the differences will be mention on P8L12 and they were not discussed further in this study, after careful consideration we decided that there is no need to emphasis them here.

Technical comments:
P2 L7: 'normally is around : : :' please replace with 'the long-term mean around 310 DU during this time : : :' It will be chaged to: where the long-term mean is about 310 DU at this time of year

P2 L10: ' : : : based on an : : :' OK

P2 L17: please refer to table 1 after the listing of the five reanalyses P2 L27: dito We do not believe that it is necessary to refer to table 1 in the introduction. We will add in P2 L27 of the original manuscript: (See table 1 for more information.)

P6 L15: please change ': : : the rest' to ': : : the other reanalyses: : :' OK

P8 L19 and P9 L19: ' : : : strong cyclonic circulation systems : : :' OK

P9 L18/19: please delete one 'because' Deleted

---

## Author Response (AR2)

**Response to Co-Editor Decision: Reconsider after minor revisions (Editor review)**

We thank the editor for his comments, our reponses are in blue.

Comments to the Author:
I think the authors address the comments from the reviewers, with the exception of the main comment from reviewer #1, which I think they only address partially. This is explained further in the comments following (I would like to see the response from the authors, but this should be a formality):

The pages and line numbers correspond to the revised manuscript.

As I see it, the authors do not quite follow the advice of referee #1 and provide a weaker statement that that suggested by the referee. I suggest the authors reword the last paragraph in sect. 6 to indicate that for studies of ozone mini-holes using reanalyses, one should follow the procedure outlined.
In our revision, we added a paragraph to that effect, which we have now changed as follows: Due to the mismatch between the mini-holes found in OMI and the mini-holes found in the reanalysis fields, **one must** paid careful attention to ensure that the regions used to study them coincide with regions where the reanalysis fields display mini-hole conditions. That is to say, it is insufficient to identify the mini-hole position in the data and then see what the reanalysis fields do at those exact locations. Rather, one **must** find the mini-holes in the reanalysis fields, compare if the events are of similar magnitudes, and if they are, study the meteorological conditions there.

I suggest the authors also provide in this section a comment on the relative merit of satellite data and reanalysis data for studying ozone mini-holes.
We will add at the end of the last paragraph in the summary: This study exemplifies the importance of assessing the reanalyses --- for which the satellite data are paramount to cover large areas --- before studying atmospheric processes and their variability.

P. 2  L. 27: Please indicate here what you will do in each section of the paper.
We will add: This paper is organized as follows: Section 2 gives an overview of the satellite data and the reanalyses used in this study. Section 3 describes the mini-hole definition used. Sections 4 and 5 show comparisons with OMI and MLS, respectively. Lastly, Section 6 provides a summary.

P. 4 L. 12: concentrations. See also caption for Fig. 2.  OK

P. 5  L. 8: I suggest you provide more details of the "flood filling" algorithm.  We will add: an algorithm that determines pixels meeting a threshold value in a 2D array.

P. 9 L. 3: reanalyses.  OK

P. 19  Fig. 4: If not too complicated to describe, I suggest the authors mention the periods over which they calculate the monthly averages.   We will add: during 2005-2014

P. 21 Fig. 8: Identify in the caption what is the reference ozone profile in panel A. Same for Fig. 9.

[revised manuscript text omitted]